# Spatially Engraving Morphological Structure on a Polymeric Surface by Ion Beam Milling

**DOI:** 10.3390/polym11071229

**Published:** 2019-07-23

**Authors:** Ansu Sun, Ding Wang, Honghao Zhou, Yifan Li, Chris Connor, Jie Kong, Jining Sun, Ben Bin Xu

**Affiliations:** 1Mechanical and Construction Engineering, Faculty of Engineering and Environment, Northumbria University, Newcastle upon Tyne NE1 8ST, UK; 2MOE Key Laboratory of Materials Physics and Chemistry in Extraordinary Conditions, Shaanxi Key Laboratory of Macromolecular Science and Technology, School of Science, Northwestern Polytechnic University, Xi’an 710072, China; 3School of Engineering and Physical Sciences, Heriot-Watt University, Edinburgh EH14 4AS, UK

**Keywords:** ion beam milling, topographic surface, wetting, contact angle hysteresis

## Abstract

Polymer surface patterning and modification at the micro/nano scale has been discovered with great impact in applications such as microfluidics and biomedical technologies. We propose a highly efficient fabricating strategy, to achieve a functional polymer surface, which has control over the surface roughness. The key development in this fabrication method is the polymer positive diffusion effect (PDE) for an ion-bombarded polymeric hybrid surface through focused ion beam (FIB) technology. The PDE is theoretically explored by introducing a positive diffusion term into the classic theory. The conductivity-induced PDE constant is discussed as functions of substrates conductivity, ion energy and flux. The theoretical results agree well with the experiential results on the conductivity-induced PDE, and thus yield good control over roughness and patterning milling depth on the fabricated surface. Moreover, we demonstrate a controllable surface wettability in hydrophobic and superhydrophobic surfaces (contact angles (CA) range from 108.3° to 150.8°) with different CA hysteresis values ranging from 31.4° to 8.3°.

## 1. Introduction

Surface patterning and modification at micro-/nano-scales have been of great importance in creating functional surfaces for a wide range of applications, such as water repelling and self-cleaning [1,2,3,4], antifouling [5], anti-icing [6], adhesion control, and drag reduction technologies [7,8]. To create surfaces with desired roughness and topography, some techniques have been commonly used, such as lithography-based plasma etching and deposition, coating on top of patterned substrates, and/or soft-lithography pattern transferring, and, more recently, creating stimuli-responsive surface cracking, wrinkling [9,10,11,12,13] and other deformations on smart material surfaces [14,15,16]. 

The focused ion beam (FIB) technique has proven its efficiency in manufacturing semiconductors, metals and metal oxides, with its unique capability for rapid prototyping and high precision [17,18]. The fundamental mechanism of FIB is that highly energetic ions driven by an electrical field knock atoms off the material surface by electro-collision and the recoil action between the ion and target material surface (Figure 1). For ion-milled surfaces, the morphological evolution can cause kinetic roughness, which has attracted increasing research interest in recent decades [19,20,21]. However, limited attempts have been reported on the topic of FIB processing on polymeric substrates, since the charging effect from the insulated polymer matrix significantly reduces manufacturing precision, and the understanding of the morphological evolution for an ion-milled polymer surface remains yet to be fully explored [22,23,24,25]. Compared to other surface morphology modification techniques, the FIB method has great potential for scalable patterning with both roughness level and geometry size ranging from a few nanometers to 10 µm.

## 2. Theoretical Background

As shown in Figure 1, ion bombardment is commonly considered as atomic processes taking place inside the bombarded material within a finite penetration depth. The ions pass through a distance *α* before they completely release their kinetic energy with a spatial distribution inside the target substrates. An ion releasing its energy at point *P* in the solid contributes energy to the surface point *O* that may induce the atoms in *O* to break their bonds and leave the surface or diffuse along it. The pattern formation by ion beam sputtering has been previously understood as the interplay between the unstable dependence of the sputtering yield on surface curvature and stabilizing surface relaxation mechanisms [26,27]. The most successful model to predict surface evolution under ion sputtering was the Bradley and Harper (BH) equation [28]. BH theory describes the ripple formation by discussing the surface topography *h*(*x*, *y*, *t*), measured from an initial smooth configuration in the (*x*, *y*) plane. However, it could not explain the surface roughening well [29,30,31]. Therefore, Makeev, Barabási and Cuerno [32] refined the noisy Kuramoto–Sivashinsky (KS) equation [33,34] based on the Sigmund theory of sputter erosion [35], where the surface was bombarded by ions, and included the Kardar–Parisi–Zhang (KPZ) nonlinear term in the BH equation. Cuerno et al [26,36,37] further developed an effective evolution equation:(1)$$\frac{\partial h}{\partial t}=-v\nabla^{2}h+\lambda_{1}{\left(\nabla h\right)}^{2}-\lambda_{2}\nabla^{2}{\left(\nabla h\right)}^{2}-K\nabla^{4}h$$
where *v*, *λ*_1_ and *λ*_2_ are the average coefficients determined by the experimental parameters such as ion flux, ion energy, etc. For an amorphous solid in equilibrium with its vapor, *K∇^4^ h* (known as the MBE equation) [38] has been studied and obtained [39,40]. Equation (1) was originally used to describe dynamic scaling on the surface under the thermal surface diffusion; here, the conditional surface diffusion factor, *K*, can be decomposed with conductivity-induced PDE constant, *D^c^* [39,41]:(2)$$K=\frac{D^{c}\beta \Omega^{2}M_{con}}{k_{b}T}\exp\left(\frac{-\Delta E}{k_{b}T}\right)$$
where *β* is the surface free energy per unit, *Ω* represents the atomic volume, *M_con_* denotes the number density of conductive particles, *k_b_* is the Boltzmann constant, *T* is the absolute temperature, and *ΔE* is the activation energy for surface diffusion. The value of *D^c^* could be determined by the evolved Nernst–Einstein equation [42,43]:(3)$$D^{c}\equiv \frac{\sigma_{dc}k_{b}T}{e^{2}M_{con}}$$

Here *σ_dc_* represents the DC conductivity of sample and *e* is the elementary charge. To simplify the discussion, the symmetric case (*δ* = *u*, which are the distribution distances in directions parallel and perpendicular along the beam) was applied to the current model, and the incident angle *θ* is zero. The linear wavelength instability could be calculated as in Ref. [34]:(4)$$l_{i}\;=\;2\;\pi {(\frac{2K}{v})}^{1/2}$$
which correlates to ion flux and matrix conductivity, and where *i* refers to the direction (*x* or *y*). Given a small incidence angle =−(Fα)/2δ, where $F\equiv \left(\epsilon Jp/\sqrt{2\pi }\right)\exp\left(-\frac{\alpha^{2}}{2\delta^{2}}\right)$ [44,45], Equation (1) could describe the surface roughening caused by PDE, after considering the conductivity induced ion diffusion by Equations (2) and (3). The *J* means the average ion flux, *ϵ* denotes the total energy carried by the ion and *p* is a proportionality constant between power deposition and rate of erosion.

The surface roughness evolution could be predicted from the following equation [26]: (5)$$\tau =v\lambda_{2}/\left(K\lambda_{1}\right)$$

Equation (5) has been applied under different experimental conditions [46,47].

We have previously demonstrated carbon-based polymer composites with exciting properties induced by enhanced electrical conductivity [48,49]. In this project, conductive polymer composites will be used to overcome the challenge from the dielectric surface charging effect during the FIB process. A new concept of conductivity-induced PDE is proposed to understand ion impacting a conductive polymer surface and to predict the surface evolution during FIB. The ion-bombarded surface topographic features with conductivity-induced PDE are theoretically predicted using Monte Carlo simulation, and also experimentally assessed. Comparative studies of FIB-induced surface patterning and morphological evolution are carried out. The emerging application of fabricated surfaces is explored with surface wetting controls. We expect that the findings in this work will advance the current understanding of FIB fabrication on polymer surfaces.

## 3. Experimental Methods

Conductive polymer nanocomposites such as polystyrene–carbon nano-particles (CNPs) were used to create the conductive polymer surfaces [50,51,52,53]. The styrene-based precursor (PS, Veriflex^®^, CRG Co. Ltd., Miamisburg, OH, US) [50,51] and the CNPs (VULCAN^®^ XC72R, CABOT, Boston, MA, US), were ultrasonically agitated in a three-neck flask for 2 h at 1000 rpm [54]. Then the curing agent (Luperox ATC50, Sigma-Aldrich, St. Louis, MI, US) was added, and the mixture stirred for 45 min. Polystyrene–carbon nano-particles composite (CNP/PS) films with a thickness of 200 µm were made by casting the mixture into a PTFE mold and baking in a vacuum oven at 75 °C for 36 h. 

Electrical conductivity was measured using an I-V testing set-up and thermo-electrical testing was performed through a Schlumberger Solartron 1250 Frequency Response Analyser from 20 to 100 °C in an isolated chamber with an ambiance of air.

A dual-beam FIB instrument (FEI, Quanta3D FEG, Thermo Fisher Scientific, Waltham, MA, US) equipped with liquid gallium ion source (Ga+, 30 KeV) and Scanning electron microscopy (SEM) was used. The topographic surface was assessed with an atom force microscopy (AFM, Triboscope, Bruker, Coventry, UK). Sputter yield was calculated through Monte Carlo simulation (TRIDYN, binary collision approximation ion irradiation simulation) [55,56], which simulates the ion irradiation of amorphous targets in the binary collision approximation. It allows for a dynamic rearrangement of the local composition of the target material [57]. Therefore, effects in high-fluence implantation, ion mixing, and preferential sputtering caused by atomic collision processes can be concluded [58]. Considering the current macromolecular-based hybrid system, an enthalpy of sublimation value (6.2 eV) was set in the simulation by consulting the chemically covalent bond energies and atom composition.

## 4. Results and Discussion

Figure 2a illustrates the conductive hybrids, based on the SEM fracture surface morphologies for 2 vol. % CNP/PS. The background SEM showed that the CNPs were distributed uniformly throughout the textured polymer matrix. The electrical current (imaginary) flowed through the conductive hybrid while the threshold network was generated. Figure 2b presents the DC conductivity results with little variations as a function of CNP concentrations at room temperature. When CNP concentrations (*φ_CNP_*) increased from 0.5 to 2 vol.%, the conductivity dramatically increased from 1 × 10^−8^ to 100 S/m, and this increment slowed down when the CNP concentration exceeded 2 vol. %. The conductivity for *φ_CNP_* > 2 vol. % was sufficient to enter the general semiconductor region.

Such a percolation network has been well understood as a polymer-based inorganic (*σ*_1_)–organic (*σ*_2_, *σ*_2_ << *σ*_1_) conducting system, or resistors and capacitors [59,60,61]. At a lower CNP concentration, conduction is mainly dominated by hopping conduction among the nanofillers, thus appearing closer to the insulator [59,61]. They became conductors when the filler concentration increased to a critical value, i.e., the percolation threshold (*φ_c_*), which formed the electron bridge within the substrate by the filler state [62]. To determine *φ_c_*, the conductivity σ was fitted based on the power laws [59,63]:(6)$$\sigma (\varphi_{\mathrm{CNP}})\propto {\left(\varphi_{c}-\varphi_{\mathrm{CNP}}\right)}^{-s}\;\mathrm{when}\;\varphi {\mathrm{CNP}}_{\;}<\;\varphi_{c}$$
(7)$$\sigma (\varphi_{\mathrm{CNP}})\propto {\left(\varphi_{\mathrm{CNP}}-\varphi_{c}\right)}^{t}\;\mathrm{when}\;\varphi {\mathrm{CNP}}_{\;}>\;\varphi_{c}$$
where *t* and *s* are the critical exponents in the conducting and insulating regions. The linear-fitting results clearly defined the threshold network with *φ_c_* = 2 vol. %, *t* = 0.858, and *s* = 4.75 (the inset in Figure 2b). Previous reports [64,65] noted that a higher critical value (*t* > 2) in a polymer/CNP system will reduce the conductive efficiency, whereas, a good conductive efficiency (*t* = 0.858) was achieved due to the uniform nanofiller distribution by the adopted techniques.

The conductivity–temperature relationship is shown in Figure 2c, where the measured conductivity gradually increased with the rising temperature, which enhanced the hopping conductivity in composites [66]. The sample conductivity for 2 vol. % CNP/PS approached the percolation limit of an insulator-dominating state, and further rises in temperature significantly increased the conductivity through the enhanced hopping. When the CNP content was above *φ_c_*, the CNP particles/clusters were more likely to link with each other, forming a continuously distributed CNP network in the matrix. Figure 2d summarizes the calculated conductivity diffusion coefficients with dependency on temperature. For the composites that hadn’t formed the threshold network, the diffusion coefficients were low, and the value was located in the ion diffusion range inside of the insulated solid (<10^−18^ m^2^/sec) [67]. With the CNP content increased, the PDE constant significantly increased from 10^−21^ to 10^−11^. It should be noticed that the diffusion constant for an ion-liquid system is 10^−11^–10^−9^ m^2^/sec [67]. This conductive network generation by adding CNP enhanced the overall ion diffusion capability dramatically. From the information in Figure 2c,d, the thermal effect on sample conductivity, or *D^c^*, which caused a changing factor of 10–100, is negligible when comparing the large improvement by increasing conductivity.

Since all coefficients in Equation (1) are determined by ion flux and *K*, the coefficient *K* can be calculated with *D^c^* (in Figure 2d) by Equations (2) and (3). We next investigate the influence from material with a fixed ion flux *Ф* = 1.2 × 10^9^ ions/(µm^2^/sec). With the Monte Carlo algorithm it could be obtained that *v* = 187 nm^2^/min, *λ*_1_ = 78.4 nm/min, and *λ*_2_ = 4373.2 nm^3^/min. The simulated roughness *τ* is displayed in Figure 3a–c, compared with the experimental AFM plots. The quantitative agreement in the order of magnitude between the experimental and the theoretical results was found for predicting the surface evolution trend, and the surface roughness decreased constantly with the CNP content increases. The experimental values were only half of the theoretical values for 1 vol. % and 2 vol. % CNP/PS composites. For 3.5 vol. % CNP/PS, the magnitude of the experimental result agreed well with theory, which could be attributed to the metallic type surface morphological evolution occurring during ion milling on the samples with high conductivity. Furthermore, the asymptotic morphologies revealed the increasing *l_i_* values as well as the reduction of *τ* with the target conductivity increases, implying a higher self-smoothing effect and a thermal relaxation mechanism led to a less defined pattern order for the hybrids. The discrepancy between the experimental data and theoretical prediction can be explained by ignoring the rapid temperature rises during ion sputter, which induces a thermal diffusion.

The milling depth values are plotted as a function of ion flux in Figure 3d–f; the grey areas represent the overall removal depth, including the targeted milling depth (500 nm), and the calculated roughness, while the up-edge indicates the accumulating value of roughness and targeted removal depth. As predicted, the self-smoothing conductive-induced PDE was found as shown in each figure. Both the experimental and numerical morphologies presented a low surface roughness associated with low ion flux values, contrary to the much higher roughness values under higher ion flux. This could be derived from the flux related parameters in Equation (5), *v*, *λ*_1_ and *λ*_2_, which change significantly with applying higher ion flux. Moreover, the experimental average removal depth was reduced at high ion flux for all samples; this could be due to the inaccurate numerical calculation at high roughness. Figure 3d–f shows the roughness reduced with increased conductivity both in experimental and numerical results, proving that the pattern characteristics are dominated by the sample conductivity. Figure 3a–c also reflects that the roughness peak at high conductivity values is broader when the sample was bombarded at the same flux values. It should be noted that the ion flux employed in this work is 10–1000 times higher than those have been reported [26,36,68]; the thermally activated surface diffusion effect could not be ignored when the target’s temperature increases, which causes the self-smoothing effect on the milled surface as well as conductivity-induced PDE does.

The experimental topographic information is summarized in Figure 4 with SEM images, AFM profiles and statistical analysis for AFM data. The deteriorating trends are presented with dependencies on ion flux and sample conductivity; the SEM observation illustrates that higher ion flux creates more surface roughness, probably combined with the re-deposition [69]. The milling precision was improved with sample conductivity increases, which could be identified from the evolving morphology in the SEM images under different ion flux; simultaneously, the AFM contour plots agree with this improvement well, with showing concentrated milled depth. The contour plots also reflect that the highest roughness appears for 1 vol. % CNP/PS, which indicates milling accuracy was lowered with low conductivity. The statistical analysis from AFM suggests wide distributed milling depths for the 1 vol. % sample especially under the high ion flux (1.25 × 10^10^ and 1.75 × 10^10^ ions/(µm^2^/sec)). Meanwhile, a concentrated distribution for 3.5 vol. % CNP/PS was observed under the low ion flux, which represented high uniformity for the milling depth. Figure 4 also reveals that the actual average milling depths were around 700 nm for most conditions with considerable errors, which some distance from the target removal depth of 500 nm. The possible reason could be the thermal induced polymer chain broken during the high energy ion sputter process, which could be understood as the thermal induced positive effect. Although improved milling precision was achieved for 2 vol. % and 3.5 vol. % CNP/PS, the actual milling depth decreases for 1 vol.% CNP/PS when ion flux increased. This could be attributed to the calculation uncertainty caused by the ultimate roughness, as previously mentioned, the residual surface charge and the re-deposition caused by molecular chain breaking [70]. Additionally, the Monte Carlo codes in this work considered the effects in high-fluence implantation, ion mixing and preferential sputtering caused by atomic collision processes, and proved a positive correspondence between ion milling efficiency and sample conductivity. However, it did not take account of the thermal induce surface diffusion, which has been previously proved with the stabilization effect on an milled surface [26,44,71].

## 5. Application Demonstration

The CNP/PS polymer matrix surfaces were FIB ion milled into different micro-roughness regions (2 × 2 mm^2^ areas pre-patterned with 20 × 20 µm^2^ square pattern arrays) with milling depths ranging from 0.5 to 1.2 µm, and *R_a_* (arithmetic mean roughness) values ranging from 700 to 4800 nm (0.7 to 4.8 µm). Different patterns are demonstrated in Figure 5a, from line array to dedicated probe shape. The processing efficiency and the precision are significantly increased. We next selected the dot array pattern (Appendix A) for the surface wetting testing. A self-assembly monolayer (SAM) of Trichloro(1H,1H,2H,2H-perfluorooctyl)silane (FOTS, Sigma-Aldrich, St. Louis, MI, US), was applied from the vapor phase at room temperature (~20 °C) for 30 min to facilitate a conformal hydrophobic layer over the CNP/PS topologies.

To set a benchmark, the static contact angle (CA) of 2 μL of deionized (DI) water on a smooth FOTS surface was measured to be 107°. Figure 5b shows that on the modified CNP/PS surface, the CA ranges from 108.3° to 150.8° (Figure 5c). Dynamic CA measurements (advancing and receding) were also taken, with different CAH (Figure 5d) values ranging from 31.4° to 8.3°. These values shown in Figure 5c,d were close to Wenzel state prediction at lower roughness (<3.5 µm), and closer to the Cassie–Baxter state at higher roughness [13] with CA = 150.8° and CAH = 8.3°, which conforms to the superhydrophobic surfaces (SHS) criteria of CA > 150° and CAH < 10°.

## 6. Conclusions

Good structure property relationships were revealed with the homogeneous dispersing state of CNPs in a PS matrix from SEM observation, measured conductivity and the stable electrical–temperature performance. The assessment of ion milled surfaces indicated that milling accuracy and surface roughness are highly dependent on the sample’s conductivity. A good agreement between experimental results and the theoretical prediction was achieved in describing the surface’s evolving trend, including the general analytical conditions for the coarsening process to occur and the roughness of the surface with different ion flux and material conductivity. Resulting micro-roughness patterns were coated with hydrophobic monolayer FOTS and demonstrated surface wettability control, resulting in hydrophobic and superhydrophobic surfaces (CA ranging from 108.3° to 150.8°) with different CA hysteresis values ranging from 31.4° to 8.3°.

It must be noted that the ion bombardment on a macromolecular surface is far more complicated than a silicon surface. In future work it would thus be interesting to investigate ion sputtering on conductive polymer (composites) surfaces with conductivity and thermal-induced PDEs, and more substrate-related factors, such as molecular chain movements and polymer degradation.

## Figures and Tables

**Figure 1 polymers-11-01229-f001:**
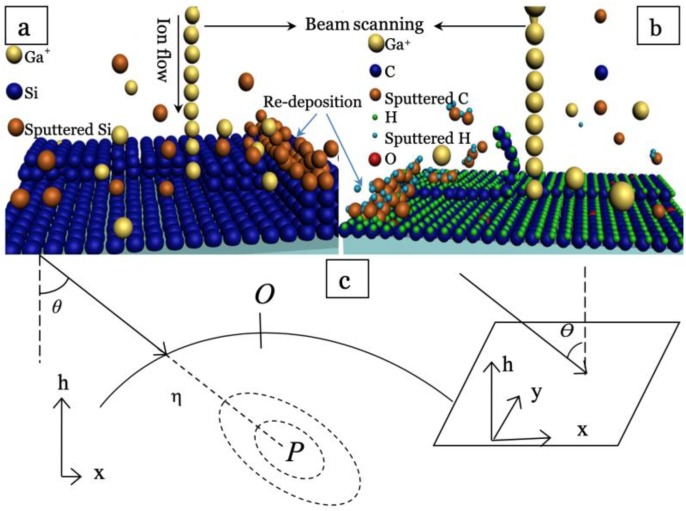
Schematics of focused ion beam (FIB) milling on (**a**) silicon and (**b**) conductive polymeric hybrid surface. (**c**) Following a straight trajectory (solid line), the ion penetrates an average distance α inside the solid (dashed line) and completely releases its kinetic energy at *P*. The dotted equal energy contours indicate the energy decreasing area around point *P*. The energy released at point *P* contributes to erosion at *O*. The inset shows the laboratory coordinate frame: the ion beam forms an angle *θ* with the normal to the average surface orientation, z, and the in-plane direction x is chosen along the projection of the ion beam.

**Figure 2 polymers-11-01229-f002:**
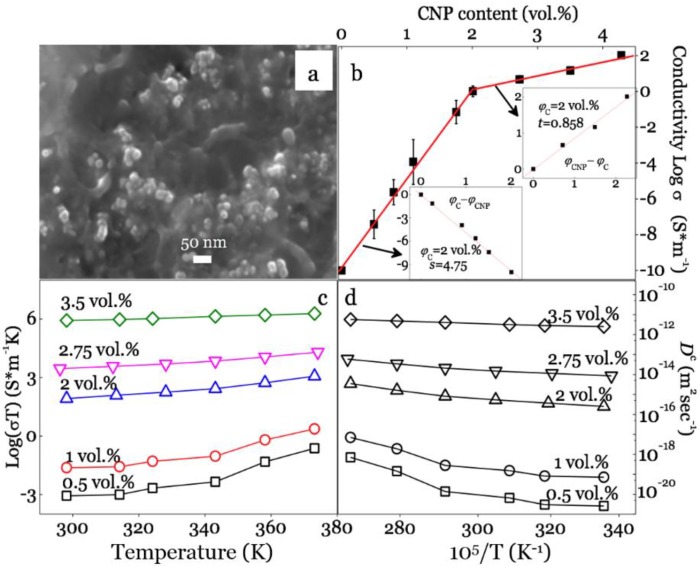
(**a**) Scanning electron microscopy (SEM) observation of the conductive surface for 2 vol. % CNP/PS; (**b**) DC conductivity results as a function of CNP content and the inset linear fitting curves needed for determining the threshold value; (**c**) DC conductivity for composites with dependency on temperature and CNP concentration; (**d**) calculated conductivity diffusion coefficients as a function of temperature.

**Figure 3 polymers-11-01229-f003:**
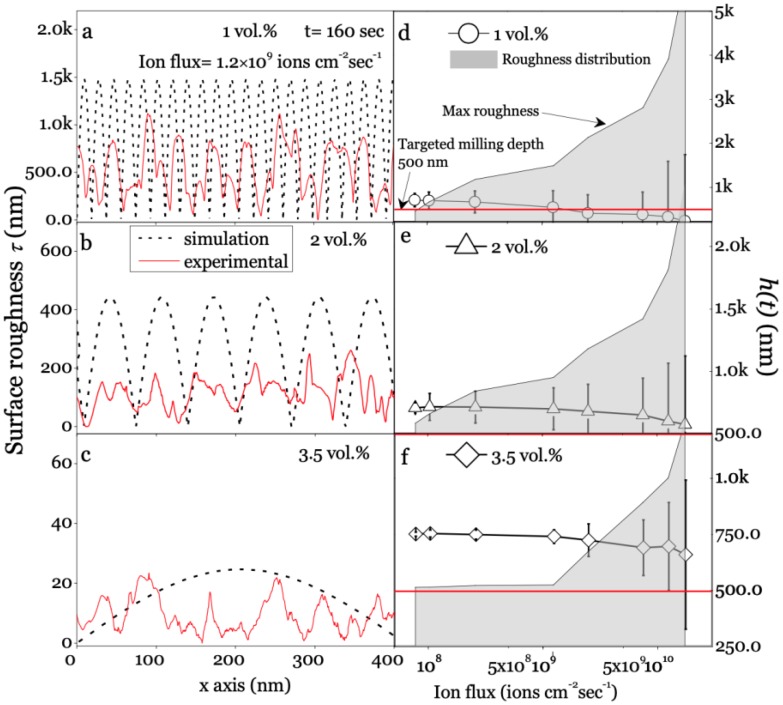
(**a**–**c**) Experimental atom force microscopy (AFM) profiles on a milled surface and numerical longitudinal plots for different composites (milling time = 160 sec); (**d**–**f**) Time evolving milling efficiency (removal depth, h(t)) with surface roughness (error bar); the gray area represents the roughness with targeted removal depth of 500 nm.

**Figure 4 polymers-11-01229-f004:**
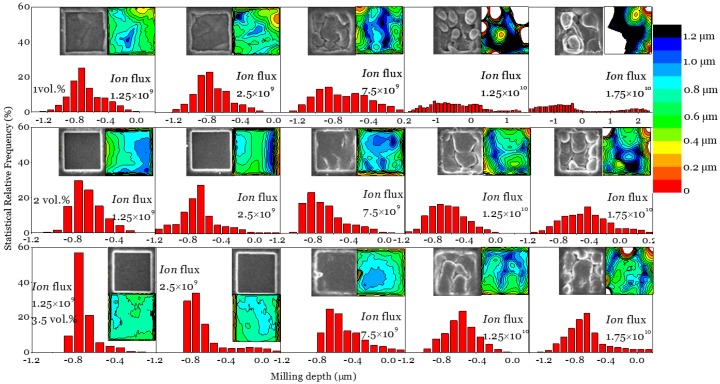
Statistical depth analysis for the milled patterns (5 × 5 µm^2^, milling time = 160 s) based on AFM results (inset, contour plots) under different ion flux, combined with the SEM images (inset).

**Figure 5 polymers-11-01229-f005:**
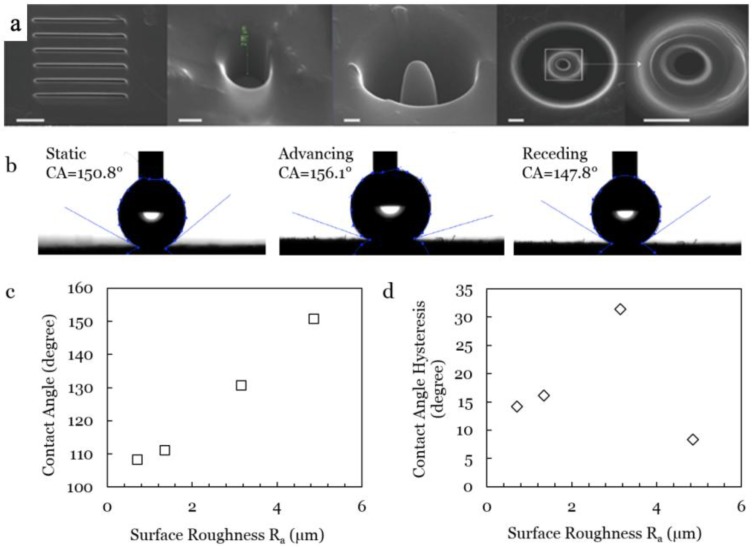
(**a**) FIB engineered nanostructures, from left to right, lines pattern, nano-hole, nano-probe, nano-tunnel; the scale bar is 500 nm. Static contact angle (CA) and contact angle hysteresis (CAH = advancing CA - receding CA) characterization on patterned CNP/PS polymer with FOTS layer. (**b**) CA and CAH values of a DI water droplet on superhydrophobic surface (Ra = 4.8 µm). Relationships between (**c**) CA and surface roughness, and (**d**) CAH and surface roughness.

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
