# Peer review of "Spatially Engraving Morphological Structure on a Polymeric Surface by Ion Beam Milling"

_polymers, 2019, doi:10.3390/polym11071229_

Round 1

Reviewer 1 Report

Dear Author,

Please check the reference no 44, which was not cited in the paper,

Thanks!

Author Response

Many thanks to the reviewer for helping us to improve the quality of this manuscript. 

This reference (now numbered 45) is now cited in line 95.  

Reviewer 2 Report

The manuscript is focused on a very interesting problem : modifying a polymer surface by polymer postive diffusion effect through FIB technology. A theoretical background is given and experimental results provided and compared to Monte-Carlo simulations results. Finally, examples are given in the field of superhydrophobic surfaces.

The résults are convincing but the manuscript is very difficult to read. I would therefore recommand the acceptation of the paper with minor revision (i.e. no extra experiments required) involving an extensive rewriting of the paper.

Some comments/suggestions to be adressed before the final acceptation :

 There seems to be a confusion between 'topography' and 'topology' (including the the paper title)

Physics underlying Eq.1 should be briefly explained

Roughness is not defined (Ra, Rq, ... ?)

Monte-Carlo simulations should also be briefly explained : what are the basis of TRYDIN ?

What are the error bars on the critical exponents s and t ?

Lines 185-187 : please explain what you mean by 'quantitative agreement' ? To my point of view, the agreement is qualitative and only quantitative in the order of magnitude

Fig. 4 : "statistical analysis frequency" is confusing

Author Response

Many thanks for helping us to improve the quality of this manuscript and recognition of our work as being very interesting.  

1.     The grammar has been improved throughout, especially the results and discussion part by co-authors who are native English speakers. – all changes are all highlighted in yellow

2.     Many thanks for the reviewer to point out these:  

a.     This has been corrected.  “Topology” is either replaced by “morphology”or “topography”.

b.     This has now been explained line 69 - 70

c.      The explanation of roughness has been added at line 248.

d.     The TRIDYN simulation has now been explained at line 129.

e.     The error bars in figure 2brefers to the conductivity results variations, now clarified in the context at line 139-140

f.      It’s now line 185, the authors agree that it’s “quantitative agreement in the order of magnitude”between the experiment and simulation results in this particular case. 

g.    Figure 4 left axistitle is now changed to “Statistical Relative Frequency”.

Reviewer 3 Report

Sun et al, report on the use of focused ion beams for surface patterning of conductive composite polymers mixed with carbon nanoparticles. Surface evolution of the polymers are studied both theoretically and experimentally. An application in surface wettability control is demonstrated using nanodot arrays. Overall, the results are technically sound. I would suggest its publication in Polymers.

1.    In Fig 3, the theoretical prediction doesn’t quantitatively match well with experimental data in terms of the surface roughness. I would suggest revising the statement in Line 185, Page 5. Also, in Fig 3, is the experimental data collected from one sample? If no, error bars should be added.

2.    Relevant reference should be included: M.W., et al, Wrinkled hard skins on polymers created by focused ion beam, PNAS, vol 104, no 4, pp. 1130-1133, 2007.

Author Response

Many thanks for helping us to improve the quality of this manuscript and recognition of our work as being technically sound.  

1.     It’s now line 188-189, we have modified the sentence to “…the magnitude of the experimental resultagreed well with theory, …”

Figure 3a-cAFM measurements have been repeated several times on different surfaces areas. However due to the nature of surface morphology measurement, the surface z-position various completely at certain “X axis” position between different scans, it would have been less helpful to include error bars in such scenarios.  

2.     Many thanks for the reviewer to point out this useful reference paper.  It is now added as No. 15.  Reference listnumbering has been adjusted accordingly.   

Reviewer 4 Report

This is an interesting work on topological structure on polymeric surface by ion beam milling. This paper can be accepted after minor revision as follows:

1.       Unique advantages of Ion Beam Milling process with respect to Topological Structure on Polymeric Surface should be critically discuss as compared to other techniques.  Why Ion Beam Milling process is selected for Topological Structure on Polymeric Surface? These two points to be discussed into the introduction section.

2.       Quality of the discussion section should be improved.

Author Response

Many thanks for helping us to improve the quality of this manuscript and recognition of our work as being technically sound.  

1.      It’s now been added in the Introduction section line 44-47:

“Compared to other surface morphology modification techniques, the FIB method has the great potential for scalable patterning with both roughness level and geometry size range from 10s nm to 10s µm. ”  

2.     Many thanks for the reviewer’s suggestion. Section 4results and discussion has now been reviewed and improved accordingly – changes are all highlighted in yellow colour.       
